# The Variance of the Polypropylene $\alpha$ Relaxation Temperature in iPP/a-PP-*p*PBMA/Mica Composites

**Jesús-María García-Martínez \*** and **Emilia P. Collar**

Polymer Engineering Group (GIP), Polymer Science and Technology Institute (ICTP), Spanish Council for Scientific Research (CSIC), C/Juan de la Cierva, 3, 28006 Madrid, Spain; ecollar@ictp.csic.es
* Correspondence: jesus.maria@ictp.csic.es

**Abstract:** By considering that the $\alpha$ relaxation related to the glass to rubber transition (obtained by dynamic mechanical analysis) of isotactic polypropylene (iPP) can be identified with the thermal history of the material (and so, with the processing step), this work deals with the changes in this transition temperature (T$\alpha$) in polypropylene/mica composites caused by the mutual effect of the other components (mica and interfacial additive). Here, the additive used is a *p*-phenylen-bis-maleamic grafted atactic polypropylene (aPP-*p*PBMA) obtained from polymerization wastes (aPP) by the authors. This additive contains $5.0 \cdot 10^{-4}$ g.mol$^{-1}$ (15% $w/w$) grafted *p*PBMA. In essence, this article has two different objectives: (1) To observe and discuss the changes in T$\alpha$ of the polymer matrix (iPP) caused by the combined effect of the other components (mica and aPP-*p*PBMA); and (2) predicting the values for T$\alpha$ in terms of both aPP-*p*PBMA and mica content for whatever composition in the experimental space scanned. This task was undertaken by employing a Box–Wilson experimental design assuming the complex character of the interactions between the components of the iPP/aPP-*p*PBMA/mica system, which define the ultimate properties of the composite.

**Keywords:** hybrid materials; compatibilizers; composites; nanocomposites; modeling; interfaces; wastes; residues; iPP; aPP

## 1. Introduction

There are three main aspects to consider when investigating polypropylene-based composites. The first point is the fact that this polymer (iPP) represents more than 60% of all the plastics materials produced in the world, remaining at the vanguard of thermoplastic-based advanced materials in terms of performance, markets, and research activities [1,2]. The second is to assume that it is the inter-phase between the components that is going to govern the ultimate properties of the material as a whole [3–6]. Third, the processing operations play a key role in the emerging morphologies, and therefore, in the generated interfacial area responsible for the final performance of the material [3,4]. This last aspect is very often forgiven in many articles in the literature, interpreting as interfacial effects the flow-dynamical variations caused by mere changes in particle size during processing. This is the reason why in this study a mineral not suffering such variations was chosen [3,4].

The definition of nano-composite by the International Union of Pure and Applied Chemistry (IUPAC) implies considering that one (or both) phase domains in the material are in the nano-metric scale at least in one dimension [7]. This definition agrees with the iPP/mica composites, with mica exhibiting one dimension (thickness) in the nano-scale order close to 30 nm, even if non-exfoliated [3,8].

One of the most efficient strategies to improve the composite properties is employing additives (the so-called interfacial agents). These additives must resemble the polymer matrix [9]. In this way, the properties of the composites depend on the combined effect of the type and the amount of the interfacial agent used [9–14]. Moreover, very often, a small amount of this additive is not only enough, but mandatory to optimize the properties of

the iPP-based composite as a whole. The interfacial agent used in this work was obtained by the authors from polymer wastes by grafting a novel functionality (p-phenylen-bis maleamic acid) to the polymer backbone instead of the usual maleic anhydride grafts, which imply several shortcomings for green chemistry and eco-friendly framework [8,15].

This type of organic–inorganic composite is a multi-component compound exhibiting at least one of the components (the polymers and/or the reinforcement) in the sub-micrometric or better in the nano-metric size domain [16,17]. It is worth mentioning that mica becomes characterized by its virtually perfect cleavage capacity at the atomic range level, conferring excellent dimensional stability to the mica/polymer composite. This mineral can be easily cleaved and exfoliated into ultra-thin flakes (with very high aspect ratios), making easier the alignment of the flakes in the matrix during the processing operations, providing a high reinforcement level [18]. Furthermore, this atomic cleavage capacity confers a high degree of hydrophobicity to the mica surface that improves the interactions at the mineral/polymer interface by the absence of traces of water (typically anchored to most silicate-based minerals.). The poor affinity between the non-polar matrix (iPP) and the polar mica particles both persists. Hence, the use of interfacial agents (usually named compatibilizers) highly improves the interactions between them [9–12,18–21]. Therefore, the chemical similarity of the interfacial agent to the polymer matrix and its affinity with the reinforcement becomes mandatory. Consequently, in the absence of an interfacial agent (as the mica and polypropylene exhibit very different polarities), the interaction at the interface is poor. As a result, the interphase between components becomes weak. Despite the vast amount of investigation conducted on the interfacial interactions, it is a fact that the precise interpretation of such complex phenomena is still open [3,8,9,11–17,22]. One of the most efficient strategies to improve the composite properties is by using additives (the so-called inter-facial agents). These additives must resemble the polymer matrix [9]. In this way, the properties of the composites depend on the combined effect of the type and the amount of the interfacial agent used [9–14]. A previous work by the authors was devoted to the influence of the components in the variation of the glass transition [4], identifying that the interfacial agent must be allocated in the amorphous phase of the system [3,4]. This aspect was confirmed by SIRM (Synchrotron Infra-Red Microscopy) spectroscopy [3,4,12,23]. Furthermore, the existence of the same critical point in terms of the component amount in the composite was found independently of the characterization method used (tensile, flexural, impact, DSC, TGA, SEM, FESEM, and so on) [3,4,8,23–25].

Focusing on the polymer matrix transitions, apart from the glass or β transition of the iPP matrix [7 °C] (related to the "free amorphous" phase of the iPP matrix able to participate in the short-range, but cooperative diffusive motions at the chain segment level) [3,4], the transition occurring between 40 and 90 °C, called the α transition, is associated with the glass-rubber like region and so rapid short-range diffusive motions are predominant [3,4,11,12,19,26,27]. These are strongly dependent on the structural parameters (mainly the molecular weight) and others related to the thermal, shear, and flow history inherent to the processing history determining the entanglement density, chain orientation, etc. [3,4]. The influence of the processing steps in the evolution of this parameter was earlier indicated by Passaglia [26] and McCrum [27].

The study of these complex systems requires experimental designs able to detect the critical points of the system. Perhaps it is well worth noting what a complex system represents, which is one consisting of many blocks able to exchange stimuli between them and the surroundings depending on the contour conditions [3,4,8,28,29]. Therefore, a route to model this type of system is using "agent-based models" [28,29]. The Box–Wilson surface response methodology for predicting Tα in terms of the components of the composite matches the "agent-based models" requirements [28–31]. This aspect has permitted the interpretation of the effect of both variables over the Tα variation from a physical viewpoint [3,4,25,28,29]. In this work, we minimized the effect of the processing step by employing the same conditions for all samples [3,4,15,28,29]. Consequently, it has been possible to isolate (as far as possible) the influence of the components in the final value

of the transition. Therefore, the objective of this study is the prediction of $T\alpha$ in terms of their components (with independence on the processing step) to better identify the end-use temperature of the system [3,4,30,31].

## 2. Materials and Methods

### 2.1. Materials

An isotactic polypropylene, iPP ISPLEN 050 (Repsol), ($\rho$ = 0.90 g/cm$^3$; $M_w$ = 334,400; $M_n$ = 59,500; $T_g$ = $-13$ °C; Tm = 164.7 °C), and phlogopite mica platelets (KMg$_3$[Si$_3$AlO$_{10}$](OH)$_2$), Alsibronz$^\circledR$ (BASF) ($\rho$ = 2.85 g/cm$^3$; specific surface BET = 1.5 m$^2$/g; and average particle larger size = 79.8 μm), were used as starting materials as received. Mica was used because its dimensional stability, mean size, and particle size distribution do not vary during the processing steps [3–6,11,12,25].

Grafted atactic polypropylene with $5 \times 10^{-4}$ mol/g $_{polymer}$ (15% *w/w*) *p*-phenylen-bis-maleamic acid grafted groups (aPP-*p*PBMA) conceived and obtained by the authors through a chemical modification process in the melt by using polymerization waste as a raw material was employed as the interfacial agent. A detailed description of the process and the characterization procedures of the grafted polymer can be read elsewhere [15]. The chemical structure of the interfacial agent used can be viewed in Figure 1.

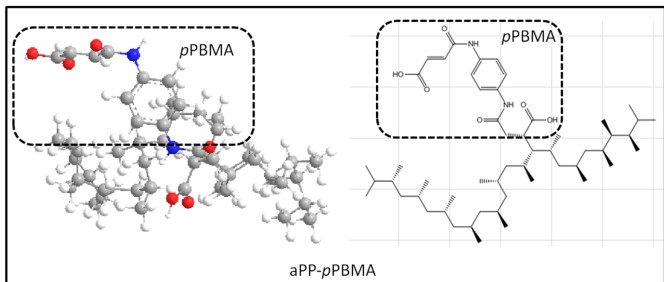

**Figure 1.** The chemical structure of the aPP-*p*PBMA interfacial agent employed in this work.

### 2.2. Sample Preparation

The composites were fabricated according to the composition compiled in Table 1. First, they were compounded in an internal mixer, then milled to obtain pellets and injected into dog-bone type 1A specimens by following the ISO 527-2 standards [32]. This process has been described in detail in a previous article [4]. Then, these specimens were machined into prismatic samples without varying the original width and thickness (19.5 × 4 × 2 mm$^3$) shaped according to the DMA test requirements [4].

**Table 1.** Experimental design and measured $\alpha$ transition temperature ($T\alpha$) according to the Box–Wilson experimental worksheet including the coded and the controlled (dosing) factors.

| | Coded Factors * | | Controlled Factors * | | | |
|---|---|---|---|---|---|---|
| **Exp** | **[Mica]** | **[aPP-*p*PBMA]** | **[Mica] (%)** | **[aPP-*p*PBMA] (%)** | **T$_\alpha$ (°C)** | **T$_\alpha$ (K)** |
| E1 | −1 | −1 | 14.4 | 1.465 | 76.9 | 350.05 |
| E2 | 1 | −1 | 35.6 | 1.465 | 64.0 | 337.15 |
| E3 | −1 | 1 | 14.4 | 8.535 | 72.6 | 345.75 |
| E4 | 1 | 1 | 35.6 | 8.535 | 85.0 | 358.15 |
| E5 | $-\sqrt{2}$ | 0 | 10.0 | 5.000 | 81.2 | 354.35 |
| E6 | $\sqrt{2}$ | 0 | 40.0 | 5.000 | 79.2 | 352.35 |
| E7 | 0 | $-\sqrt{2}$ | 25.0 | 0.001 | 69.0 | 342.15 |
| E8 | 0 | $\sqrt{2}$ | 25.0 | 9.999 | 78.3 | 351.45 |
| E9 | 0 | 0 | 25.0 | 5.000 | 73.0 | 346.15 |
| E10 | 0 | 0 | 25.0 | 5.000 | 73.5 | 346.65 |
| E11 | 0 | 0 | 25.0 | 5.000 | 75.2 | 348.35 |
| E12 | 0 | 0 | 25.0 | 5.000 | 74.1 | 347.25 |
| E13 | 0 | 0 | 25.0 | 5.000 | 71.1 | 345.25 |

* $x_1$ = [Mica]; * $x_2$ = [aPP-*p*PBMA].

### 2.3. Characterization

The real particle content in the composite after being processed (and not merely the nominal) was checked by thermo-gravimetric analysis (TGA) on a TAQ50 apparatus equipped with automatic sample feeding. About 20 mg of each sample was placed in a crucible and heated from 30 up to 750 °C at a heating rate of 10 °C/min under a nitrogen atmosphere (90 mL/min). Figure 2 shows the TGA curves for all the experiments performed in this work, and these data are compiled in Table 2.

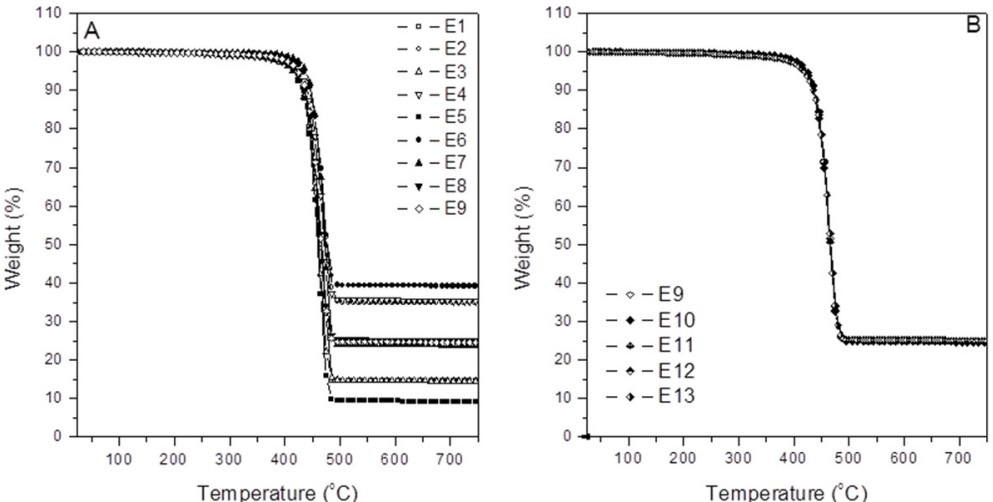

**Figure 2.** TGA spectra for the indicated samples. (**A**) Different composite compositions. (**B**) Replicas.

**Table 2.** The nominal mica content of each experiment in the Box–Wilson model versus the real content as determined by the residue in TGA (Figure 2).

| Exp | Nominal Mica Content (%) | Real Mica Content (%) |
|---|---|---|
| E1 | 14.4 | $14.5 \pm 0.1$ |
| E2 | 35.6 | $34.9 \pm 0.2$ |
| E3 | 14.4 | $14.5 \pm 0.1$ |
| E4 | 35.6 | $35.4 \pm 0.3$ |
| E5 | 10.0 | $9.8 \pm 0.3$ |
| E6 | 40.0 | $39.6 \pm 0.5$ |
| E7 | 25.0 | $24.9 \pm 0.2$ |
| E8 | 25.0 | $25.0 \pm 0.1$ |
| E9 | 25.0 | $24.8 \pm 0.3$ |
| E10 | 25.0 | $24.7 \pm 0.3$ |
| E11 | 25.0 | $24.8 \pm 0.2$ |
| E12 | 25.0 | $24.9 \pm 0.2$ |
| E13 | 25.0 | $25.0 \pm 0.1$ |

As it can be appreciated, all the composites leave a residue that is almost coincident with the nominal [Mica]. This indicates an excellent control of the dosing. This is a key aspect of the further discussion in a robust manner on the change of properties in the composite. This basic aspect is too often ignored in several works in the literature.

The particle dispersion and particle distribution were observed by environmental scanning electron microscopy (ESEM) over cryogenically fractured samples [25] by using a Philips XL30 ESEM scanning electron microscope (15 kV) over gold-coated specimens by a Thermo VG Scientific SC7640 sputter. Figure 3 shows some images of the fractured samples.

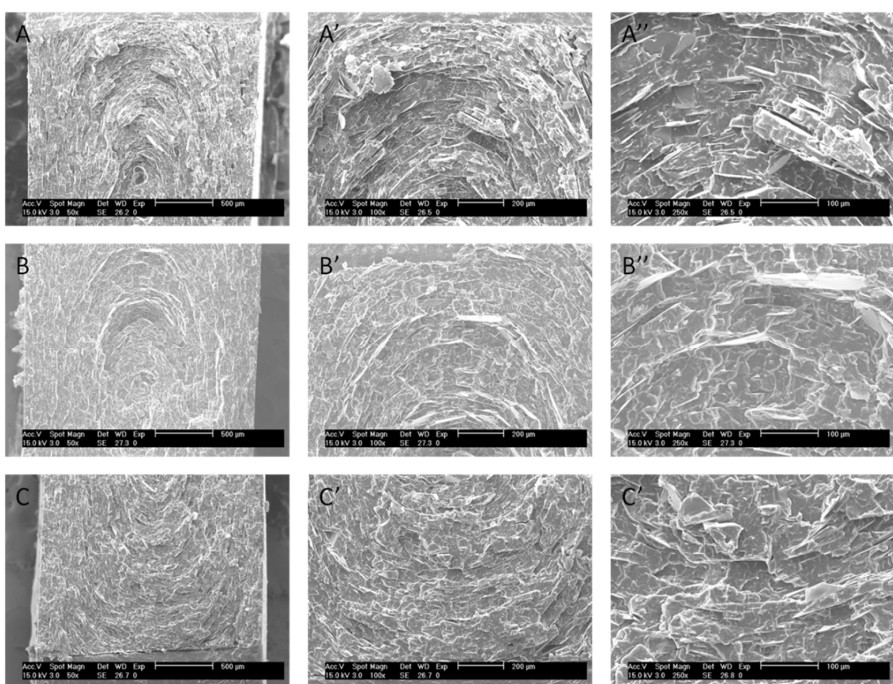

**Figure 3.** FESEM micrographs on the IPP/mica 75/25 composite fracture surfaces. Sample E7 (**A**,**A′**,**A″**); sample E8 (**B**,**B′**,**B″**); and sample E9 (**C**,**C′**,**C″**). Magnification (50×, 100×; and 250×).

A dynamic mechanical analyzer, DMA, (METTLER DMA861) under tension mode was used to obtain the DMA spectra. Therefore, the ASTM D5026 standards and recommendations were followed for the measurements. This implies that the dynamic mechanical parameters were measured within the range of linear viscous–elastic behavior of the material (12N oscillating dynamic force applied at a 1 Hz fixed frequency, 3 μm amplitude, and 2 °C/min heating rate). The temperature was varied in the 40 °C to 140 °C intervals to detect the α transition. A series of additional samples were run into the −20 °C to 140 °C range to merely visualize all the transitions in this interval (α and β in Figure 4). The election of the 1 Hz frequency is reported to be the best option to detect the interactions at an interface level [4,8,26,29]. Both the rather low frequency and displacement applied to the samples help to avoid whatever nonlinear behavior and any morphological changes are provoked by eventual internal heat generation.

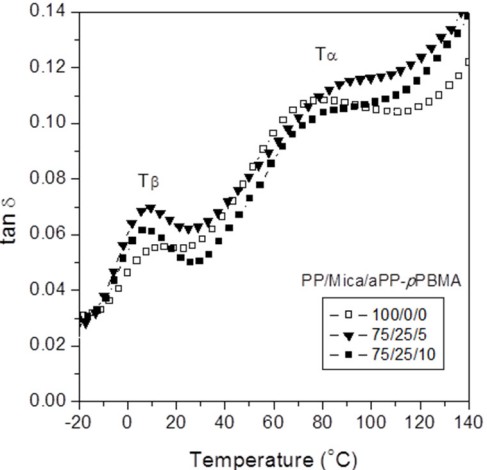

**Figure 4.** Evolution of the damp factor with temperature and the α and β transition for a pristine iPP and two composites with different amounts of aPP-*p*PMBA.

### 2.4. Mathematical Model

A two independent variable Box–Wilson statistical experimental design (sDOE) was used to model and predict the $\alpha$ transition temperature of the iPP/aPP-pPBMA/mica composite system within the experimental space scanned [30,31]. At a glance, this methodology consists of a central rotary composite design with some experiments ($2^k$ + 2k + 1) augmented with (2 + k) replicated runs in the central point coded as (0,0), wherein k is the number of independent variables (here mica and aPP-pPBMA) [30,31]. A fuller description of the model as used here can be read in our previous article [4]. Thus, the data for $T\alpha$ measured from the DMA spectra compiled in Table 1 were fitted to a quadratic polynomial that describes the property evolution along all the compositional ranges studied. These data are listed in Tables 3 and 4.

**Table 3.** Statistical parameters and coefficients of the polynomials (Polynomial Equation: $a_0 + a_1 \cdot x_1 + a_2 \cdot x_2 + a_3 \cdot x_1 \cdot x_2 + a_4 \cdot x_1^2 + a_5 \cdot x_2^2$) *.

|  | $\langle r^2 \rangle$ (%) | LF (%) | CF (%) | Linear Terms | | | Interaction Term | Quadratic Terms | |
|---|---|---|---|---|---|---|---|---|---|
|  |  |  |  | $a_0$ | $a_1$ | $a_2$ | $a_3$ | $a_4$ | $a_5$ |
| $T\alpha$[K] | 94.86 | 14.9 | 99.7 | 377.7 | −2.104 | −2.741 | 0.1688 | 0.02441 | −0.004234 |

* $x_1$ = [Mica]; $x_2$ [aPP-*p*PBMA].

**Table 4.** Confidence coefficient (%) and *t*-values for the different terms of the model obtained for $T\alpha$ (Polynomial Equation: $a_0 + a_1 \cdot x_1 + a_2 \cdot x_2 + a_3 \cdot x_1 \cdot x_2 + a_4 \cdot x_1^2 + a_5 \cdot x_2^2$).*.

|  | Linear Parameters | | Interaction Parameter | Quadratic Parameters | |
|---|---|---|---|---|---|
|  | $x_1$ | $x_2$ | $x_1 \cdot x_2$ | $x_1^2$ | $x_2^2$ |
| $T\alpha$[K] | 7.136 (99.9%) | 3.710 (99.1%) | 7.909 (99.9%) | 4.528 (99.5%) | 0.872 (57.1%) |

* $x_1$ = [Mica]; $x_2$ = [aPP − *p*PBMA].

## 3. Results and Discussion

### 3.1. TGA Analysis

The correct dosing of the mineral in the composites was checked by TGA by evaluating the remaining residue, as shown in Figure 2.

### 3.2. SEM Observations

In this work, morphological observations have just been used to ascertain the particle distribution and dispersion in the composite. Thus, to merely observe this aspect, a series of SEM micrographs obtained from the fracture surface for the E7, E8, and E9 samples is shown in Figure 3. These samples were chosen as they corresponded to the critical amount of mica (25%) previously determined elsewhere [3,4,24,25].

Otherwise, the observation of these images allowed us to conclude that the particles were very well dispersed in the matrix by following a concentric pattern. This pattern is imposed by the developing flow elements during the injection molding process. It is noteworthy to mention that the mica particles dispersed in the iPP bulk may be considered as stress concentrators. Moreover, it was observed that some differences in the morphology were found depending on the presence of aPP-*p*PBMA in the iPP/mica/aPP-*p*PBMA composite. Thus, a sharper fracture surface can be distinguished in the case of the sample with a scant amount of interfacial agent (E7). Meanwhile, the fracture surface corresponding to samples with more aPP-*p*PBMA, exhibited a much more homogeneous and smoother fracture, evidencing the effect of this interfacial agent.

Even with such a low magnification, the so-called core-shell morphology typical to injection-molded specimens can be identified jointly to the three classical flow regions: shear flow; elongation flow (both close to the shell region); and the fountain flow around the core symmetrically arranged through the section of the observed sample.

### *3.3. Dynamic-Mechanical Analysis*

3.3.1. Background

Before discussing the T$\alpha$ variation, a series of concepts must be clarified, where at least four different concerns must be considered.

The first is related to the interactions occurring between the species involved. In previous works published by the authors, a model describing the possible interactions at the inter-phase between the mineral particles, the polymer matrix, and the interfacial agent was discussed [3,4]. However, a brief discussion of these is presented to aid better comprehension of further discussions on the variation in the T$\alpha$ in this work. In essence, the model considers two diverse types of amorphous fractions in the iPP matrix [3,4]. One serves as the interconnecting material between the crystallites of the IPP crystalline macro-aggregates. The other is allocated between the reinforcement and the iPP crystalline phase surrounding it [3,4]. This permits matrix continuity in the system. This latter is strongly affected by the processing and molding conditions. Furthermore, the allocation of the interfacial agent (of amorphous character) into the "free amorphous" fraction of the matrix affects the transitions. This was already reported for the glass transition [4], since the "free amorphous fraction" is highly constrained by imbibing the mica particles. This fact is expected to decrease at the $\alpha$ transition due to the effect of higher temperature in the polymer chain dynamics.

Nothing but as a resume, in this model, a simplification of the complex scenario that takes place at the interface is appreciated [4]. In essence, we can define some zones between the iPP crystalline and the inorganic phase: one between the crystal and the amorphous phase of the iPP matrix. Here, an amorphous/crystal inter-phase contains the unordered (amorphous) sequences of the isotactic polypropylene jointly with some atactic sequences of the iPP (nor fully isotactic as a whole). At this point, we must point out that the inorganic phase (mica) must be mandatorily embedded in the amorphous phase of the system [3,4,8,10,18]. This implies that an amorphous/mica interface is present [3,4]. Additionally, in between the iPP lamellae and mica, the amorphous phase of the system (consisting of tie segments and other segments excluded from the crystal) is also identified. Moreover, and due to the grafted groups also becoming excluded from the crystalline domains, the interfacial agent must be hosted majorly in this area [3,4,8,10,18]. This is precisely accurate in our case since the interfacial agent (aPP-*p*PBMA) reflects a mainly amorphous character, being so mandatory allocated in this phase [3,4,7,8,10–13,15,19,24]. From the latter, it must be concluded that the presence of both mica and the interfacial agent in the amorphous domains must disrupt this one. This exerts a significant influence on the glass transition of the matrix since the interaction balance between the species involved (and so to the mobility of the amorphous phase) is varied [3,4]. The latter remains an aspect that is equally influenced on the $\alpha$ transition, but more softly due to the less constrained scenario favored for higher temperatures [3].

Second, replacing a minor fraction of iPP with aPP-pPBMA is found to greatly enhance the interactions across the dynamic interface between the polymer and the reinforcement [3,4,7,8,10–13,15,19,24]. This fact implies the existence of a critical value that depends on the type and amount of the interfacial agent used and on the processing history of the material (a core concept very often not even considered in the literature) [3,4,10,15,24]. Since a study of the effect of the interfacial agent in a thermoplastic-based composite has been performed under the same processing operations, we had two options: to employ a constant amount of interface agent with a varied number of grafts, or to incorporate a different amount of the interfacial agent by keeping the grafting degree constant. The last is the route we followed in the present study [3–6,10,15,24].

The third concern to bear in mind is the desirability of the flow-elements' preferential orientation, providing a significant degree of alignment to the mica platelets. Moreover, the morphological variations of the mineral and the eventual changes in the particle size and size distribution caused by the processing steps may be falsely assigned to modifications of the inter-phase. Therefore, it is mandatory to indicate that the inorganic platelets used in

this work (mica) have been demonstrated not to suffer significant changes in particle size and particle size distribution during the processing step [3,4,8,23–25]. Additionally, the real and precise mica content in the material must be reported to not identify as interfacial modifications those due merely to changes in the flow-dynamic of the system (and are then hardly traceable). Nevertheless, since the platelets are mainly aligned with the applied force, the percolation point provides a reinforcing effect by facilitating its transmission through the polymer matrix [3–6,8,11,24].

Particularly in our case, a fourth aspect to consider was the phenomena related to this transition (the $\alpha$ transition) of the polymer matrix (iPP). In the fact, this is a transition usually identified between 40 °C and 90 °C [1,6–8,22–24] (in our case between 62.08 °C and 84.28 °C). This is associated with rapid short-range diffusive motions sharply dependent on the molecular mass and the chain entanglement density. This transition is explained by the polymer rearrangements occurring in the crystalline domains; across the amorphous/crystal inter-phase, and does not exclusively depend on the amorphous/crystal ratio but on how both phases are connected. Additionally, this transition is strongly dependent on the processing steps and more, if this provides the material of an orientation of the flow elements (and to the platy reinforcement in it) [1,4–6,26,27].

### 3.3.2. Dynamic Mechanical Spectra: Determination of T$\alpha$

In this work, we discussed the $\alpha$ transition variations as determined by the loss factor (tan $\delta$) from the DMA spectra for the iPP/mica composite and how this parameter is affected by the combined effect of both the mineral and the interfacial modifier used. Thus, the evolution of the tan $\delta$ with temperature for all the samples of the Box–Wilson worksheet in Table 1 is visualized in Figure 5. For the objective of this work, we used the values for the $\alpha$ transition temperature. Typically, this transition appears between 40 °C and 90 °C [3,4,8,9,11,15,24,25].

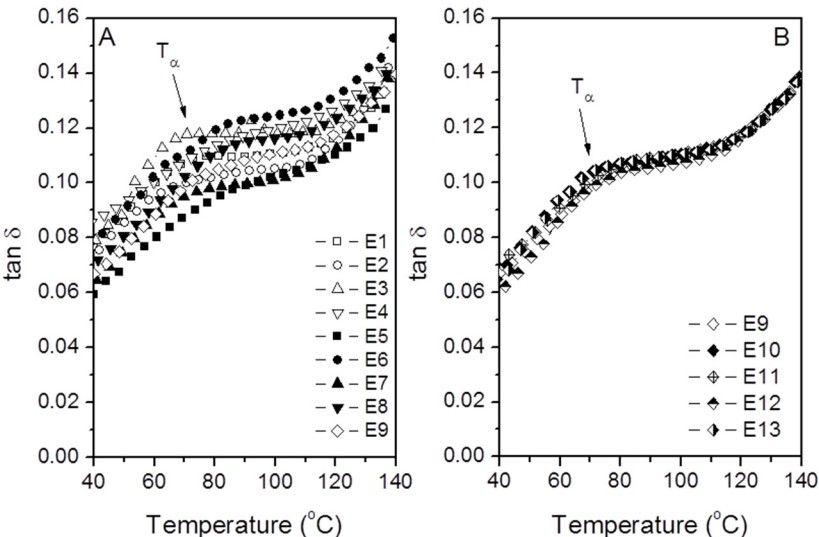

**Figure 5.** Evolution of the damp factor with temperature and the $\alpha$ transition for the indicated samples. (**A**) Different composite composition. (**B**) Replicas.

In our case, it varied from 64.0 °C (sample E2) up to 80.5 (sample E4) with the same amount of mica (35.6%) and different interfacial agents (1.46 vs. 8.53%) (Table 1). This provides a preliminary idea on the great effect of aPP-pPBMA in the system. Otherwise, the whole set of values for this $\alpha$ transition temperature has been compiled in Table 1. As mentioned hitherto, a maximum difference of 16.5 °C was identified between sample E4 and E2 (same mica content) by varying only the amount of interfacial agent, suggesting that aPP-pPBMA greatly affects the $\alpha$ transition value. In contrast to what occurred to the

glass transition, the $\alpha$ transition of the neat iPP showed values in between those observed in Table 1.

The value for the pristine iPP T$\alpha$ was around 70 °C, as measured by DMA under the same conditions [8] (Figure 4). It is important to mention that we found a positive maximum difference between the neat iPP and the highest filled compound (E4) of 10.5 °C, and a maximum negative for E2 of 6 °C (with the same filler amount but different interfacial agent content), which suggest that the value for such a transition of the composite depends on the combined effect of both the filler and the agent.

Figure 5 shows the evolution of the loss factor with the temperature of each of the composites in Table 1. At a glance, all of them followed a similar pattern. Figure 5B shows the evolution of the replicas, and a high reproducibility was concluded. Additionally, these figures show that the dissipation capabilities of the iPP matrix did not suffer any abrupt change at all, conversely to what happened with the glass transition [4]. Nevertheless, we reported elsewhere [3,4,8,9,11,12] that T$\alpha$ for the 90/10 iPP/Mica pristine composite was around 74 °C. In Table 1, we observed that the presence of 5% of aPP-pPBMA (E5) increased this value by 7.2 °C (81.2 °C). In the same way, if we compare the T$\alpha$ value for the 65/35 iPP/mica pristine composite (T$\alpha$ = 63.3 °C) [3,4,8,9,11,12] with E2 and E4, higher values due to the interfacial agent interactions were detected. Even more strenuous, when the neat 75/25 iPP/mica composite (T$\alpha$ = 42 °C) was compared with those with aPP-pPBMA, we could see that the presence of traces of the interfacial agent implied a great increase in this temperature (E7; T$\alpha$ = 69 °C).

The above described inform on the remarkably complex scenario defined previously wherein the mica platelets disrupt the polymer bulk (being 25% a critical value), providing a fraction of the polymer segments to be ordered [3,4,8,9,11,12]. Consequently, the amorphous region trapped in the iPP/mica interface coating the mica particles appears constrained. Therefore, when the sample is heated, a relaxation occurs, but is influenced by the amorphous nature of the aPP-pPBMA interacting enough to have a strong influence on the final value of such an $\alpha$ transition. In essence, the presence of aPP-pPBMA in these regions plays a twofold effect. On one hand, and due to its amorphous character, it introduces more amorphous phases into the system. Otherwise, this constrains grafted groups interacting with the mica platelets and so aids in the inter-connectivity of the system if a certain filler threshold is reached. On the other hand, the interfacial agent becomes more mobile with temperature and therefore, if the aPP-pPBMA is efficient in interacting with the mica domains, the T$\alpha$ must increase. What is clear is that this emerging complex scenario influences the T, where this transition appears in the spectra, the influence of each of the components in isolation not being so evident. Hence, the use of Box–Wilson methodology was revealed as a powerful tool to study this complex system.

### 3.4. Polynomial Fits and Analysis of Variance (ANOVA)

The values for T$\alpha$ for the experimental design followed are listed in Table 1. Therefore, these values were fitted to a quadratic model through Box–Wilson surface response methodology [30] to obtain a polynomial equation describing the evolution of the $\alpha$ transition temperature. As a result, Table 3 compiles the polynomial obtained with the confidence factor ($<r^2>$), jointly with the ANOVA (analysis of variance), through the lack of fit (LF) and the confidence factor (CF) coefficients. Thus, we observed a value for $<r^2>$ equal to 94.86%, which is excellent for a quadratic model since values higher than 75% for this parameter are considered as a very good fit [30,31]. Additionally, Table 3 includes the "lack of fit" (LF), a parameter linked to the percentage of pure error due to any factors dismissed by the model but significant. Accordingly, we obtained a value of LF equal to 14.9%, indicating that only this value may explain other factors ignored by the model, possibly the rough approximation of considering the processing step as the same, despite the different composition. Conversely, the very high value of the confidence factor (CF = 99.7%) indicates the proper significance of the independent variables chosen to model the T$\alpha$ evolution of the composite system for the entire experimental space scanned.

Consequently, the parameters in Table 3 robustly confirmed the choice of studying the iPP/aPP-pPBMA/mica composite system using the Box–Wilson forecasts. However, the limitations of the model must be checked. Therefore, we included scattering plots for the predicted versus the measured $\alpha$ transition temperature in Figure 6, where we could see a very good correlation between them.

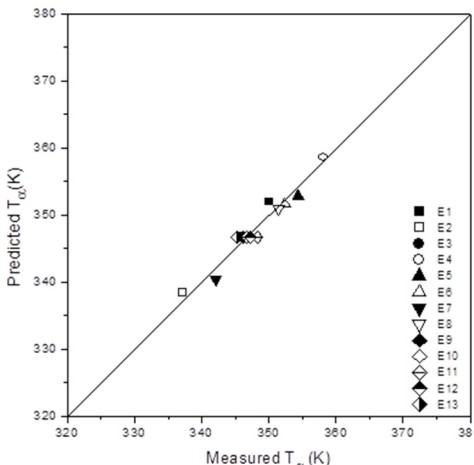

**Figure 6.** Measured versus predicted values of the $\alpha$ transition temperature (T$\alpha$).

Following the model parameters, Table 4 compiles the t-values (and the % confidence coefficient) for each of the terms of the polynomial linking T$\alpha$ with the mica and aPP-pPBMA amounts in the composite. Immediately, the very high significance levels for each of the parameters in the polynomial, except the quadratic term for aPP-pPBMA, were observed. Thus, by keeping in mind that the more influence and significative terms are those *t*-values higher than two [30,31], we see that it is the reinforcement and the interfacial agent, and the interaction between them has a great influence on the final values for T$\alpha$. Thus, the *t*-value equal to 7.136 for [Mica], 7.2, for [Mica]$^2$, 4.528 for [aPP-pPBMA], and 7.909 for the interaction terms implies that the significance (and so the real influence in the final value) was above 99% in all the cases.

All of the above-mentioned agrees with the disruption capability of mica caused by being surrounded by the amorphous phase with the interfacial agent (with an amorphous character) [3,4]. Consequently, the combined effect of mica and aPP-pPBMA is a prime aspect to consider in the T$\alpha$ evolution of the system since the confidence coefficient was close to 100% (*t*-value = 7.907) [3,4,8].

### 3.5. The Influence of the Composite Composition in the T$\alpha$ as Determined by DMA Spectra

It is important to remark that, based on the data in Table 1, not the greater amount of interfacial agent and mica content present in the composite, but a higher $\alpha$ temperature was reached. The latter indicates that the behavior of the iPP/mica system becomes complex, and so the effect of the presence of the interfacial agent is not evident. This information is very often missed when undertaking investigations utilizing random experiments. Consequently, the critical values for the amount of mica and aPP-pPBMA due to their interactions with each other, rather than in isolation, strongly affect the behavior of the $\alpha$ transition evolution of the iPP matrix in the composite, as concluded from other properties [3,4,8,10,15,24,25]. Hereinafter, the evolution of the $\alpha$ transition temperature of the iPP phase in the composite as a function of the reinforcement and the interfacial agent can be studied and discussed based on the Box–Wilson model forecasts.

Figure 7 plots the contour map of the $\alpha$ transition of the iPP matrix as a function of the content of mica and interfacial agent, and an inverse saddle evolution was observed. This means that optimal coordinates in the experimental space scanned were identified [30]. In our case, we observed a soft valley for values in the 15% up to 35% of mica regardless

of the amount of aPP-pPBMA. In fact, between 15% and 35% of mica, the values of the isolines were constant for such a reinforcement concentration. Therefore, determining the real influence of the interfacial agent was not so evident.

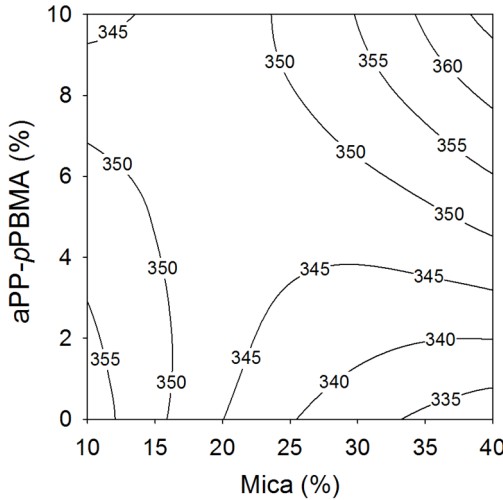

**Figure 7.** Evolution of Tα (K) with the mica and aPP-*p*PBMA contents.

With the objective of a better understanding of the Box–Wilson predictions, we included two parametric plots wherein the critical points are shown. In this way, Figure 8 represents the evolution of Tα with the mica content for the indicated [aPP-pPBMA]. It is interesting to observe that below 20–25% of mica, the higher the interfacial agent amount, the lowest variations in Tα were found. The latter indicates that below this threshold, an excess of aPP-pPBMA plays a counter-productive effect on this transition, possibly due to the excess of amorphous material that an overdosed aPP-pPBMA may cause on the generated morphologies due to the processing operations. However, the tendency was just the contrary, with increasing amounts of mica beyond this threshold. Thus, the higher the aPP-pPBMA, a higher variation of Tα can be detected. What is clear is that the critical point in the system is located in the 20–25% mica interval, regardless of the quantity of interfacial agent used. This implies that the reorganization capabilities of the crystal/amorphous balance of the iPP are stable enough to retain the fingerprint after the processing steps [3,4].

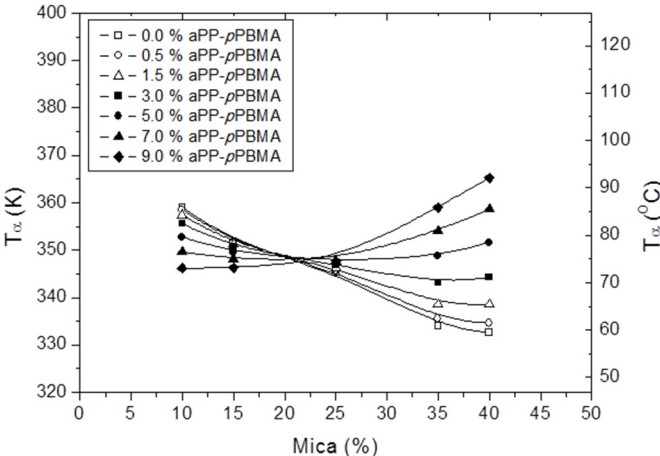

**Figure 8.** Evolution of Tα with the mica content at interfacial agent dosing.

Similarly, Figure 9 follows the evolution of Tα with [aPP-pPBMA] for different contents of mica. At a glance, two families of curves were observed: one with negative slopes represented the ones with a lower content of mica (below 20%) and the other with positive

slopes for [Mica] higher than 20%. This second family showed increasing values of Tα with the interfacial agent content. One spot to remark is that the first family of curves exhibited almost constant values until [aPP-pPBMA] close to 3%. This exactly matches the optimum for the interfacial agent concluded from the mechanical properties (tensile, flexural, and impact) [10,23,24]. Therefore, this transition is strongly influenced by the processing step [25,27], which seems to be evidenced by this fact [9,18]. Additionally, the second family presented a dual behavior depending on the interfacial agent content for values of mica above 20%. Therefore, for values below [aPP-pPBMA] = 3%, the higher the mica, the lower the Tα, and vice-versa.

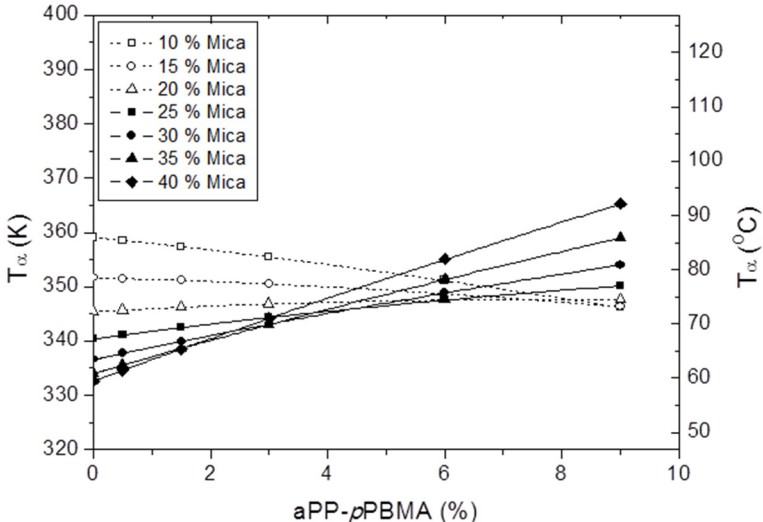

**Figure 9.** Evolution of Tα with the aPP-pPBMA content at the indicated amount of reinforcement.

This fact indicates that above 20% of mica, the effect of the interfacial agent is to increase the Tα. This effect is magnified when the increase in mica is more strenuous. Thus, this fact may be explained in terms of the model described in the background section. Conversely, the tendency once surpassed this critical point (3% aPP-pPBMA), which is just the contrary. Therefore, the higher the mica content, the higher the Tα with an increasing interfacial agent.

In summary, the 20–25% mica coordinate was identified as one of the critical points of the system. Since the change in the slope of the curves produced was close to 3% aPP-pPBMA, this may be considered a critical coordinate. These same critical points have been detected in studies of highly processing-step dependent mechanical properties (tensile, flexural, impact) and have been published elsewhere [10,23,24].

## 4. Conclusions

In this work, the evolution of the α transition temperature of the polymer matrix in iPP/mica/aPP-pPBMA was studied and modeled by the Box–Wilson experimental methodology. The use of this DOE (Box–Wilson) has been demonstrated to be a powerful tool in the study, prediction, and physical interpretation of the system. It is shown that the interaction between the components is what governs the ultimate behavior of the system, and so the value of the α transition temperature. The fact that a greater amount of each component does not lead to a more outstanding level of interactions has been concluded. The mineral content of 20–25% and interfacial agent content close to 3% were proven to be the critical coordinates for the Tα of the matrix in the iPP/mica/aPP-pPBMA system. It is important to remark that these critical points are the same as those obtained for the mechanical and thermal properties.

Moreover, this fact provides information on the linking of this threshold parameter (Tα) not only with the processing and thermal history fingerprint, but on the components present in the composite system. The real possibility of predicting the Tα provides the

researcher with the option of designing materials with a desired upper end-use temperature. This information, jointly provided for other properties, is the key in designing materials with a priori known performance.

**Author Contributions:** Both authors contributed equally to this article. All authors have read and agreed to the published version of the manuscript.

**Funding:** The results discussed in this present work were partially obtained under the auspices of the MAT 2000-1499 and MAT2013-47902-C2-1-R Research Projects.

**Institutional Review Board Statement:** Not applicable.

**Informed Consent Statement:** Not applicable.

**Data Availability Statement:** Not applicable.

**Conflicts of Interest:** The authors declare no conflict of interest.

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
