# Peer review of "The Variance of the Polypropylene α Relaxation Temperature in iPP/a-PP-pPBMA/Mica Composites"

_jcs, doi:10.3390/jcs6020057_

Round 1
Reviewer 1 Report
REVIEWER COMMENTS__ Variance of the polypropylene α relaxation temperature in iPP/a-PP-pPBMA/Mica composites.
General Comments
The work is of some interest. However, the presentation of the work and use of language must be improved. The authors struggle to present their thoughts and arguments with sufficient clarity. Manuscript appears to be written in an informal style that is inconsistent with scientific communication. The article was written in the active voice instead of a passive voice and with repetitive and inappropriate use of “we” and “the latter”. Significant sections of the manuscript will need to be rewritten.
In the light of an earlier publication in POLYMERS journal on the same material [García-Martínez, J. M., & P Collar, E. (2020). On the Combined Effect of Both the Reinforcement and a Waste Based Interfacial Modifier on the Matrix Glass Transition in iPP/a-PP-pPBMA/Mica Composites. Polymers, 12(11), p.2606. https://doi.org/10.3390/polym12112606 ] authors should provide justification why the current manuscript should no be considered a duplicate publication.
In doing this authors should define in the manuscript with citation their definition of Tg and Ta, implications and importance of these in the system(s) under study, and also clarify the classification of the polymers; i.e. symmetric or unsymmetric, and amorphous or crystalline. Hence this work should be placed in context of authors earlier work with visible citations to this earlier study and the reason for the present study and its addition to previous knowledge.
Suggestions to address some of the errors are presented below under specific comments.
Specific Comment
LINE 09: Delete “here”.
LINE 10: Write “isotactic” instead of “the isotactic”.
LINE 11: Indicate the significance of Ta.
LINE 25: Write “consider” instead of “take in mind”.
LINE 33-35: Sentence is not clear, re-phrase to improve clarity.
LINE 35: Delete “one”.
LINE 37-40: Sentence is not clear, re-phrase to improve clarity.
LINE 43-47: Sentence is too long and lacks clarity. Please, re-phrase to improve clarity.
LINE 47: Write “small” instead of “scant”.
LINE 48-49: Delete “To mention that”.
LINE 49: Write “in this work” instead of “here”.
LINE 51: Write “usual” instead of “usually”.
LINE 53: Write “organic” instead of organism.
LINE 53: Delete “revealed as”
LINE 54: Write “materials” instead of “compounds”.
LINE 54: Write “in which at least one of the components” instead of exhibiting at least one of the components”.
LINE 56: Delete “well”.
LINE 55-56: Delete “At this point,”
LINE 58: Delete “Decidedly,”.
LINE 60: Delete “, And so,”. Join the phrase to the sentence.
LINE 61: Write “Furthermore,” instead of “Further”.
LINE 64-66: Rephrase to improve clarity. Sentence lacks clarity.
LINE 68: Write “persists” instead of “both stays”.
LINE 69: Delete “As”.
LINE 70: Write “the chemical similarity of the interfacial agent to the polymer” instead of “the chemical resembling of the interfacial agent with the polymer”.
LINE 71: Delete “real”.
LINE 71-74: Verbose sentence, Rephrase to improve clarity. Sentence lacks clarity.
LINE 77-80: Rephrase to improve clarity. Sentence lacks clarity.
LINE 80-82: Rephrase to improve clarity. Sentence lacks clarity.
LINE 77-89: Authors keep talking of “latter” in this section in a confusing way. This paragraph should be re-written with improved clarity.
LINE 90-99: It is necessary to illustrate the DMA scan showing plethora of transitions in this material and present as in-set what you are referring as the α-transition.
LINE 90: State the glass transition temperature of the material in brackets to give the reader some perspective.
LINE 102: Delete “So,”
LINE 104-127: Re-phrase this section in a style consistent with scientific communication (not informal communication).
LINE 134: Presented where? In which Table?
LINE 134-136: Rephrase to improve clarity. Sentence lacks clarity.
LINE 137: Delete “A”.
LINE 151: Include citation of the ISO standard mentioned.
LINE 158: Delete “So,”
LINE 168-172: Rephrase to improve clarity.
LINE 168: Write “processing” instead of “processed”.
LINE 175: Insert citations for the experimental design mentioned.
LINE 195: Delete “Previously”.
LINE 197: Delete “occurring”.
LINE 198-237: Re-phrase this section in a style consistent with scientific communication. Lacks clarity in the present form. Check the confusing and repetitive use of “the latter”.
LINE 243-266: Re-phrase this section in a style consistent with scientific communication. Lacks clarity in the present form.
LINE 269-273: Sentence is too long and lacking in clarity.
LINE 273: Which values? Please clarify.
LINE 282: Write “Contrary” instead of “Conversely”
LINE 283: “of the neat up shows values in between the observed in table 2.” Rephrase this. It lacks clarity.
LINE 285: Write “important” instead of “substantial”.
LINE 288: Write “which” instead of “what”.
LINE 291-292: Insert “(a)” and “(b)” as is appropriate in the title of Figure 2, so that the reader is not confused.
LINE 296-298: Rephrase to improve clarity. Sentence lacks clarity.
LINE 300-301: Rephrase this : “we observe that just the presence of 5% of aPP-pPBMA (E5) increases 7.2 ºC this value up to 81.2ºC.”
LINE 301,304,333,337,352,359,362,392,398: Do not use “We” as have been used repeatedly in this manuscript. Write in the passive tense. Effect this correction all through the manuscript
LINE 301-333: Rephrase to improve clarity. Sentence lacks clarity.
LINE 335: Delete “this”.
LINE 337: Delete “enough”.
LINE 340: Write “high” instead of “extraordinary”.
LINE 350: Write “using” instead of “through”.
LINE 353: Write “good” instead of “superb”.
LINE 366-372: Rephrase to improve clarity. Sentence lacks clarity.
LINE 378-382: Rephrase to improve clarity. Sentence lacks clarity.
LINE 382: Write “undertaking” instead of “ubdertaken”.
LINE 384: Write “with each” instead of “each”.
LINE 389: Delete “So,”
LINE 391: “the latter”? That do you mean by this. Rephrase!
LINE 392-395: Rephrase to improve clarity.
LINE 400: Rephrase; “lets visualize the evolution of Tα with the mica content for the indicated amount”.
LINE 401-445: Re-phrase this section in a style consistent with scientific communication. Lacks clarity in the present form.
LINE 449: Delete “of”.
LINE 450: Delete “here”.
LINE 452: Delete “that is”.
LINE 453-464: Re-phrase this section in a style consistent with scientific communication. Lacks clarity in the present form.
Reviewer 2 Report
- The Introduction section is too long compared to the entire article.
- The authors should provide more detailed information about suppliers/producers of the materials used in the research.
- Section 2.2. The authors refer to Table 1. However, Table 1 in section 2.4 does not apply to the compositions of the tested materials. Add the missing table.
- The TGA and SEM test parameters must be included in the Characterization section. The reference to other articles is insufficient.
- There are no references in the text to Figure 3 and Figure 6. Moreover, the Authors refer to Figure 7, which is not included in the article. This should be clarified.
- Conclusion section should be better supported by research results in accordance with the JCS journal requirements.
- The English language of the article should be checked by a Native Speaker.
Round 2
Reviewer 1 Report
REVIEWER COMMENTS__ROUND_2_The variance of the polypropylene α relaxation temperature in 2 iPP_a-PP-pPBMA_Mica composites
General Comments:
The quality of the manuscript has been markedly improved. In addition the authors have presented satisfactory responses/rebuttals to issues highlighted earlier.
Few minor corrections to be addressed are presented below under specific comments.
Specific Comments
Line 45 & 74: “remoter properties” Please rephrase, this is ambiguous.
Line 113: Add units to the average particle size.
Line 114: Write “dimensional stability” instead of “dimension stability”.
Line 114: Write “do not” instead of “don’t”.
Line 226: Write “interactions occurring” instead of “occurring interactions”.
Line 229-230: Write “However, a brief discussion of these are presented to aid better comprehension of further discussions about the variation of the Tα in this work.” instead of “Therefore, it is perchance worth resuming here this one generate an easier further discussion about the variation of the Tα.”.
Line 233: Delete “And”.
Line 458: Delete “in the way”.
Line 463: Write “In summary” instead of “In the resume”.
Line 466: Write “in studies of” instead of “when studied”.
Line 474: Write “is what” instead of “is which”.
Line 479: Write “as that” instead of “as the”.
Author Response
Dear reviewer.
After yours, our answers have been listed:
Reviewer Comments:
General Comments:
The quality of the manuscript has been markedly improved. In addition the authors have presented satisfactory responses/rebuttals to issues highlighted earlier.
Authors’ Answer:
Thank you very much for your general comments. We appreciate them indeed.
Reviewer Comments:
Few minor corrections to be addressed are presented below under specific comments.
Authors’ Answer:
Under yours, an answer is provided.
Specific Comments
Line 45 & 74: “remoter properties” Please rephrase, this is ambiguous.
Ok. Done. Marked in pink in the manuscript.
Line 113: Add units to the average particle size.
Ok. Done. Marked in pink in the manuscript.
Line 114: Write “dimensional stability” instead of “dimension stability”.
Ok. Done. Marked in pink in the manuscript.
Line 114: Write “do not” instead of “don’t”.
Ok. Done. Marked in pink in the manuscript.
Line 226: Write “interactions occurring” instead of “occurring interactions”.
Ok. Done. Marked in pink in the manuscript.
Line 229-230: Write “However, a brief discussion of these are presented to aid better comprehension of further discussions about the variation of the Tα in this work.” instead of “Therefore, it is perchance worth resuming here this one generate an easier further discussion about the variation of the Tα.”.
Ok. Done. Marked in pink in the manuscript.
Line 233: Delete “And”.
Ok. Done. Marked in pink in the manuscript.
Line 458: Delete “in the way”.
Ok. Done. Marked in pink in the manuscript.
Line 463: Write “In summary” instead of “In the resume”.
Ok. Done. Marked in pink in the manuscript.
Line 466: Write “in studies of” instead of “when studied”.
Ok. Done. Marked in pink in the manuscript.
Line 474: Write “is what” instead of “is which”.
Ok. Done. Marked in pink in the manuscript.
Line 479: Write “as that” instead of “as the”.
Ok. Done. Marked in pink in the manuscript.
Thank you very much for your superb revision work.
Reviewer 2 Report
Article can be published in present form.
Author Response
Thank you for recommending to accept the article in its present form.